# RMSL: Weakly-Supervised Insider Threat Detection with Robust Multi-sphere Learning

## Abstract

Insider threat detection aims to identify malicious user behavior by analyzing logs that record user interactions. Due to the lack of fine-grained behavior-level annotations, detecting specific behavior-level anomalies within user behavior sequences is challenging. Unsupervised methods face high false positive rates and miss rates due to the inherent ambiguity between normal and anomalous behaviors. In this work, we instead introduce weak labels of behavior sequences, which have lower annotation costs, i.e., the training labels (anomalous or normal) are at sequence-level instead of behavior-level, to enhance the detection capability for behavior-level anomalies by learning discriminative features. To achieve this, we propose a novel framework called Robust Multi-sphere Learning (RMSL). RMSL uses multiple hyper-spheres to represent the normal patterns of behaviors. Initially, a one-class classifier is constructed as a good anomaly-supervision-free starting point. Building on this, using multiple instance learning and adaptive behavior-level self-training debiasing based on model prediction confidence, the framework further refines hyper-spheres and feature representations using weak sequence-level labels. This approach enhances the model's ability to distinguish between normal and anomalous behaviors. Extensive experiments demonstrate that RMSL significantly improves the performance of behavior-level insider threat detection.

## 1 Introduction

Nowadays, modern information systems have become indispensable core components in the operation of enterprises and organizations, with various monitoring data such as user behavior records continuously generated by these systems. **Insider Threat Detection** (ITD) (Silowash et al., 2012; Costa et al., 2016; Alzaabi & Mehmood, 2024) typically aggregates these regards into behavioral sequences for analysis, aiming to automatically identify anomalies. By detecting such anomalies, organizations can promptly recognize potential threats and take proactive measures to prevent losses.

However, current studies (Yuan et al., 2019; 2020; Vinay et al., 2022; Le & Zincir-Heywood, 2021a; Le et al., 2020; Zheng et al., 2022; Tuor et al., 2017; Wang et al., 2021) primarily focus on sequence-level detection, and there is insufficient research on fine-grained behavior-level detection. Given that a behavior sequence might consist of hundreds or thousands of behaviors, identifying specific anomalous behaviors can significantly help reduce the cost of manual screening and localization, making it highly significant. This paper primarily investigates **behavior-level ITD**.

Dealing with the behavior-level ITD task presents several unique challenges. **The first challenge** is the scarcity of behavior-level annotations. Due to the extreme rarity and stealthiness of anomalous behaviors, it is impractical to provide anomaly annotations for such a large number of behaviors. Almost all ITD studies (Du et al., 2017; Shen et al., 2018; Wang et al., 2021; Ni et al., 2025) train unsupervised or single-class models to learn normal patterns and identify behaviors that deviate from these patterns as anomalies. However, there still are some problems in real-world scenarios where it's impossible to enumerate all normal patterns during training, and the boundaries between normal and abnormal are blurred, leading to high false positives and miss detection rates. Introducing supervised signals regarding anomalies can help the model effectively distinguish between normal and abnormal. To address the first challenge and strike a trade-off between annotation costs and improving detection performance, this work explores a weakly-supervised setting for behavior-level detection by

introducing only some sequence-level annotations as inexact supervision, named **Weakly-supervised ITD** (WITD), as shown in Figure 1.

The cost of obtaining sequence-level annotations is relatively lower, as it only requires labeling whether a rough interval contains anomalies. Moreover, as more and more systems begin to integrate AIOps (Gulenko et al., 2020), the avenues for obtaining sequence-level annotations have become more diversified. Once an anomaly in monitoring metrics or a system failure is captured, the approximate time range of the anomaly occurrence can be given.

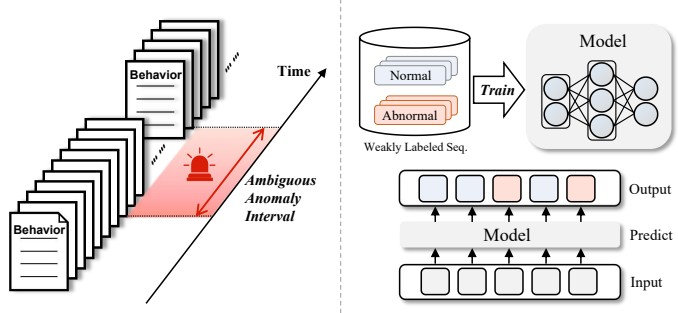

Figure 1: Illustration of Weakly-Supervised Insider Threat Detection.

**The second challenge** is how to efficiently utilize easily accessible normal data to appropriately model the normal patterns of data. DeepLog(Du et al., 2017) and TIRESIAS(Shen et al., 2018) learn to predict the next behavior given the historical behavior sequence and detect behaviors that deviate from the model's prediction as anomalies. OC4Seq(Wang et al., 2021) learns to compress all normal data into single minimal volume hyper-sphere and detects anomalies by predicting the distance of the input from the center of the hyper-sphere. In this paper, we argue that assuming normal data follow a unimodal distribution (i.e., all normal data can be contained within a hyper-sphere) is inappropriate for the ITD task. In the real world, using a single hyper-sphere may not adequately describe all normal patterns. To provide different descriptions for different normal patterns, we propose Robust Multi-sphere Learning (RMSL). In RMSL, we use multiple hyper-spheres to represent different normal patterns of behaviors and determine anomalies by combining classification separability with the degree of deviation from the hyper-spheres.

We designed a three-stage progressive training strategy to optimize the model for obtaining robust representations: the multiple hyper-spheres based zero positive warm-up stage, the multiple instance learning stage, and the adaptive behavior-level self-training debiasing stage. In the first stage, we optimize the model using only normal behavior sequences without any anomalous positive examples, i.e., the zero positive scenario. This provides a good unsupervised starting point for anomaly detection, enabling the model to have some predictive anomaly scoring ability. Subsequently, to enhance the anomaly detector's ability to explicitly distinguish between normal and anomalous behaviors, we refine multiple hyper-spheres and feature representations by using sequence-level annotations as weak supervision in the second stage. This naturally transitions to WITD, making the detector more robust, which is highly beneficial for tasks such as detecting subtle disguised anomalous behaviors in the field like insider threat detection. Some studies(Feng et al., 2021; Lv et al., 2023) have shown that multiple instance learning (MIL) exhibits a certain degree of selection bias. After the second stage, we further divide behaviors based on the model's confidence in the third stage, and propose a progressive adaptive behavior-level self-training method to learn more robust representations.

The contributions of this paper are as follows:

- We propose a novel weakly supervised learning framework, Robust Multi-sphere Learning (RMSL), to address the challenge of label scarcity in behavior-level anomaly detection. To the best of our knowledge, we are the first to formulate the fine-grained behavior-level insider threat detection problem in the context of MIL.

- We develop a multiple hyper-spheres based anomaly detector with three-stage progressive training: starting from a zero-positive initialization and gradually incorporating sequence-level supervision to enhance the model's ability to distinguish between normal and anomalous behaviors.

- Extensive experiments on CERT r4.2 and r5.2 datasets demonstrate state-of-the-art performance, achieving 9.78% and 3.98% AUC improvements over 16 baselines on the two datasets, respectively.

## 2 PROBLEM DEFINITION

Given a set of weakly labeled behavior sequences $\mathcal{D}_L = \{S^{(i)}, Y^{(i)}\}_{i=1}^{|\mathcal{D}_L|}$ as the training set $\mathcal{D}_{train}$, where each behavior sequence $S^{(i)} = \{e_l^{(i)}\}_{l=1}^{N_S^{(i)}}$ contains $N_S^{(i)}$ behaviors, $e_l^{(i)}$ denotes the $l$-th behavior in the sequence $S^{(i)}$, and $Y^{(i)} \in \{0, 1\}$ is a sequence-level label. An anomalous behavior is denoted as $e_l^+$, while a normal behavior is denoted as $e_l^-$. If a sequence contains at least one anomalous behavior, it is considered an anomalous sequence, denoted as $S^+$. Otherwise, the sequence is considered a normal sequence, denoted as $S^-$. The goal of WITD is to learn a mapping function $f(\cdot|\cdot; \theta)$ using the weakly labeled behavior sequences in the training set $\mathcal{D}_{train}$, which generates an anomaly score $f(e_l^{(i)}|S^{(i)}; \theta) \in \mathbb{R}$ for each behavior $e_l^{(i)}$. If the anomaly score of a behavior exceeds a detection threshold $\tau_a$, it is classified as an anomalous behavior, where $\theta$ represents the parameters of the model. The model performance is evaluated using a test set $\mathcal{D}_{test} = \{S^{(i)}, \mathbf{Y}^{(i)}\}_{i=1}^{|\mathcal{D}_{test}|}$ with behavior-level labels, where $\mathbf{Y}^{(i)} = \{y_l^{(i)}\}_{l=1}^{N_S^{(i)}}$ and $y_l^{(i)} \in \{0, 1\}$ is a behavior-level label.

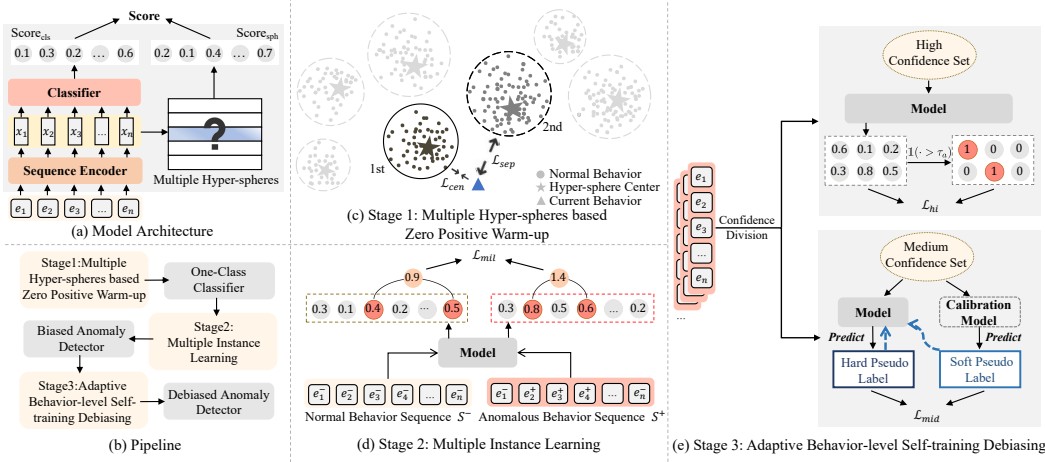

Figure 2: Overall architecture of RMSL.

## 3 METHODOLOGY

To address WITD, we propose Robust Multi-sphere Learning (RMSL) to detect whether a certain behavior $e_l$ in the given behavior sequence $S$ is anomalous. The overview of RMSL is depicted in Figure 2. The model architecture of RMSL consists of three components: a sequence encoder, multiple hyper-spheres based normal prototypes, and an anomaly classifier. The optimization of RMSL employs a progressive training strategy divided into three stages to obtain robust representations: the multiple hyper-spheres based zero positive warm-up stage, the multiple instance learning stage, and the adaptive behavior-level self-training debiasing stage.

### 3.1 MODEL ARCHITECTURE

In this subsection, we provide a detailed description of the architecture of RMSL. It consists of three components and ultimately produces behavior-level anomaly scores.

**Sequence encoder**. We first utilize a sequence encoder to generate the representation $\mathbf{x}_l$ of the behavior, which includes the behavior embedding and the sequence context encoding process. Specifically, we project the behavior code $e_l$ into an embedding space using an embedding layer, obtaining the embedding vector $\mathbf{e}_l$:

$$\mathbf{e}_l = \text{Embedding}(e_l). \tag{1}$$

Subsequently, we need to encode the contextual information in the behavior sequence to obtain the representation of the entire behavior sequence. GRU(Chung et al., 2014) is a widely used architecture

that effectively captures temporal dependencies between elements in the sequence through the gating mechanisms. For the behavior sequence encoder, our bidirectional GRU employs two layers:

$$\mathbf{X} = (\mathbf{x}_1, \mathbf{x}_2, \dots, \mathbf{x}_{N_S}) = \text{GRU}(\mathbf{e}_1, \mathbf{e}_2, \dots, \mathbf{e}_{N_S}), \qquad (2)$$

where $\mathbf{x}_l$ represents the contextual representation of behavior $e_l$.

**Multiple hyper-spheres based normal prototypes**. To address the challenge of appropriately modeling the normal patterns of data, some previous works such as Deep SVDD(Ruff et al., 2018), DeepSAD (Ruff et al., 2019) and OC4Seq(Wang et al., 2021) utilized a minimal-volume hyper-sphere to encapsulate these normal patterns by compressing all normal data into it. However, in real-world scenarios, considering normal behaviors as originating from a multi-modal distribution is more appropriate. In this work, we do not use a single hyper-sphere to store the normal, which is insufficient to uniformly describe all normal patterns. Instead, we employ $M$ learnable hyper-spheres as prototypes to store and memorize different normal patterns, and optimize these hyper-spheres to create compact representations of diverse underlying distributions in the data, naming it multiple hyper-spheres based normal prototypes. The centers of these hyper-spheres are denoted as $\mathbf{p}_m \in \mathbb{R}^d$ ($m = 1, \dots, M$), where $d$ is the feature dimension. For each behavior $e_l$, given its contextual representation vector $\mathbf{x}_l$, we can compute the distance $d_{l,m}$ from this behavior to each hyper-sphere center. A larger distance from the behavior to the center of its nearest hyper-sphere implies that the behavior is dissimilar to all historical normal patterns and is more likely to be an anomaly. This process can be formulated as follows:

$$d_{l,m} = \|\mathbf{x}_l - \mathbf{p}_m\|_2, \quad 1\text{st} = \arg \min_{m=1}^{M} d_{l,m}, \quad Score_{\text{sph}}(e_l|S) = \|\mathbf{x}_l - \mathbf{p}_{1\text{st}}\|_2, \qquad (3)$$

where $d_{l,m} \in \mathbb{R}^{N_S \times M}$, $1\text{st} \in \{m\}_{m=1}^{N_S}$ denotes the index of the nearest hyper-sphere to behavior $\mathbf{x}_l$, and $Score_{\text{sph}}(e_l|S) \in \mathbb{R}$ represents the deviation score of the behavior relative to the hyper-spheres, named as hyper-spheres based deviation score.

**Anomaly classifier**. We use a discriminative anomaly classifier $\mathcal{M}_{cls}$ to predict whether a behavior belongs to the anomaly class. This classifier consists of a self-attention layer(Vaswani et al., 2017) and a fully connected layer. The self-attention layer further enhances the representation ability of behavior features, yielding the representation $\mathbf{x}_l^{\text{attn}}$, and introduces additional parameters to better adapt to the classification task. The fully connected layer is used for the final classification decision. The entire anomaly classifier can be formulated as:

$$Score_{\text{cls}}(e_l|S) = \text{sigmoid}\left(\mathbf{w}_{FC}^{\top}\mathbf{x}_l^{\text{attn}} + b_{FC}\right), \qquad (4)$$

where $\mathbf{w}_{\text{FC}} \in \mathbb{R}^d$ is the weight matrix, $b_{\text{FC}} \in \mathbb{R}$ is the bias, $sigmoid(\cdot)$ is the sigmoid activation function, and $Score_{\text{cls}}(e_l|S)$ reflects the score of behavior $e_l$ being classified into the anomalous class from the perspective of classification separability.

**Anomaly scores**. In this work, we do not use a single classifier to generate anomaly scores but instead contribute anomaly scores from complementary perspectives. The anomaly classifier provides discriminative scores based on class separability, while the hyper-spheres based deviation score quantifies the degree of deviation of behaviors from normal patterns. This dual-scoring mechanism enables a more comprehensive assessment of anomalies. For the behavior $e_l$ in sequence $S$, we define the total anomaly score as:

$$f(e_l|S; \theta) = \alpha \times Score_{\text{cls}}(e_l|S) + (1 - \alpha) \times Score_{\text{sph}}(e_l|S), \qquad (5)$$

where $\alpha \in [0, 1]$ is a hyperparameter, which we refer to as the dual scoring balance factor.

## 3.2 Stage 1: Multiple Hyper-spheres based Zero Positive Warm-up

In this stage, we constructed a one-class classifier based on multiple hyper-spheres in the zero positive scenario (i.e., optimizing the model using only normal behavior sequences without any anomalous positive examples) as a good anomaly supervision-free starting point to warm up for the second stage multiple instance learning. This is based on the consideration that, at the beginning of training, the model is not well-trained yet, and the anomaly score prediction function does not have a clear mapping relationship. Directly selecting the behaviors with the highest anomaly scores from the anomaly sequence might not be truly anomalous, causing errors during the early phases of

multiple instance learning optimization. These errors will accumulate as the model trains, leading to suboptimal performance. Using the one-class model as a starting point can improve the model's ability to select anomalous samples in the early phases of multiple instance learning. Specifically, we use two losses to constrain the optimization of hyper-spheres.

**Multi-Center loss**. We extend the standard center loss (Wen et al., 2016) from multi-class to multi-spheres. For a normal sequence $S^-$, we minimize the distance of each behavior in the sequence to its nearest hyper-sphere, such that behaviors of the same normal pattern cluster into corresponding compact hyper-spheres in the feature space:

$$\mathcal{L}_{cen} = \frac{1}{N_{S^-}} \sum_{l=1}^{N_{S^-}} \|\mathbf{x}_l - \mathbf{p}_{1\text{st}}\|_2^2, \tag{6}$$

where $N_{S^-}$ denotes the length of the normal sequence $S^-$.

**Hyper-sphere separability loss**. Using only the multi-center loss may lead to learning meaningless results, such as hyper-sphere collapse, where all centers of hyper-spheres are optimized to converge to a single point, losing the significance of storing normal patterns in multiple hyper-spheres. To encourage different hyper-spheres to reflect different normal patterns, we propose a soft hyper-sphere separability loss that enforces the distance between a behavior and the second nearest hyper-sphere center to be greater than the distance to the nearest hyper-sphere center, thereby increasing the distance between different hyper-spheres to ensure separation between hyper-spheres representing different patterns:

$$\mathcal{L}_{sep} = \frac{1}{N_{S^-}} \sum_{l=1}^{N_{S^-}} \text{BCE}\left( \frac{\exp(\|\mathbf{x}_l - \mathbf{p}_{2\text{nd}}\|_2)}{\exp(\|\mathbf{x}_l - \mathbf{p}_{1\text{st}}\|_2) + \exp(\|\mathbf{x}_l - \mathbf{p}_{2\text{nd}}\|_2)}, 1 \right), \tag{7}$$

where 1st and 2nd are respectively the indices of the nearest and second nearest hyper-spheres to $\mathbf{x}_k$, $2\text{nd} = \arg \min_{m=1, m\neq 1\text{st}}^{M} d_{l,m}$, and $\text{BCE}(\cdot, \cdot)$ is used to calculate binary cross-entropy losses.

The total training loss at this stage can be calculated as $\mathcal{L}_1 = \mathcal{L}_{cen} + \lambda_{sep}\mathcal{L}_{sep}$. After training, the model tends to have a smaller deviation score $Score_{\text{sph}}$ for normal behaviors, while anomalous behaviors, which the model has not seen, are likely to be further from the hyper-spheres storing the normal behavior patterns. Consequently, their $Score_{\text{sph}}$ are also more likely to be larger than that of normal behaviors. We use this property to help warm up MIL in the next stage.

### 3.3 STAGE 2: MULTIPLE INSTANCE LEARNING

To address the behavior-level ITD task, our ultimate goal is to ensure that the anomaly scores for anomalous behaviors are higher than those for normal behaviors, effectively separating out the anomalies. Typically, achieving this goal requires relying on detailed behavior-level annotations for model optimization. However, with the Multiple Instance Learning (MIL) technique(Carbonneau et al., 2018; Wu et al., 2015; Zhang et al., 2013), we can adopt a more efficient approach: consider the sum of the anomaly scores of the highest-scoring behaviors within a sequence as the anomaly score for the entire sequence. Based on this, ensure that the anomaly score of an anomalous sequence is higher than that of a normal sequence, thereby enabling the optimization of the model using sequence-level labels. This method indirectly achieves the goal of scoring anomaly behaviors higher than normal behaviors, which can be formalized as:

$$\sum_{l\in\Omega_{S^+}} f(e_l|S^+;\theta) > \sum_{l\in\Omega_{S^-}} f(e_l|S^-;\theta), \tag{8}$$

where $\Omega_{S^+}$ and $\Omega_{S^-}$ represent the indices of behaviors with the highest anomaly scores in the anomalous sequence $S^+$ and the normal sequence $S^-$, respectively. Therefore, in the second stage of the progressive training strategy, we introduce sequence-level weak supervision signals. By applying the MIL technique, based on the one-class model obtained in the first stage, we further enhance the model's ability to distinguish whether a behavior is abnormal or not. By selecting the behaviors with the highest anomaly scores $f(e_l|S^{(i)};\theta)$ within a sequence (i.e., $\Omega_{S^{(i)}} = \max_{e_l\in S^{(i)}} \left( f(e_l|S^{(i)};\theta) \right)$), and minimizing the binary cross-entropy loss $\mathcal{L}_{mil} = \frac{1}{|\mathcal{D}_L|} \sum_{i=1}^{|\mathcal{D}_L|} \text{BCE}(\hat{Y}^{(i)}, Y^{(i)})$ using the sequence-level labels, the entire model $\mathcal{M}$ is optimized to further refine feature representations and hyper-spheres, where $\hat{Y}^{(i)} = \frac{1}{|\Omega_{S^{(i)}}|} \sum_{l\in\Omega_{S^{(i)}}} f(e_l|S^{(i)};\theta)$.

### 3.4 STAGE 3: ADAPTIVE BEHAVIOR-LEVEL SELF-TRAINING DEBIASING

After obtaining an anomaly score prediction model $\mathcal{M}$ through MIL in the second stage, $\mathcal{M}$ acquires an initial capability to distinguish anomalies. However, due to the mechanism of MIL that optimizes the model by selecting only a few representative behaviors, there exists a prediction bias. In the third stage, we use adaptive behavior-level self-training debiasing technology to fully utilize the information of all behaviors in the sequence. By generating efficient pseudo labels to optimize the model while introducing minimal noise, we eliminate the prediction bias and improve anomaly detection performance. The debiased model is named as $\mathcal{M}'$, and the corresponding parameters are denoted as $\theta'$.

Specifically, based on the model trained in the second stage, we calculate the confidence of the model's prediction for each behavior in the sequence $S$. Monte Carlo (MC) Dropout(Gal & Ghahramani, 2016) provides a good way to estimate it. It treats the network parameters $\theta$ as random variables following some distribution $q(\theta)$. By using the dropout operation(Hinton, 2012) during each forward pass, we can approximate sampling from the distribution of the model parameters. Through multiple forward passes, we can approximate the distribution of the model's predictions. The expectation and variance of the distribution of the anomaly score of a behavior $e_l$ can be estimated from the mean and variance of the outputs generated by multiple forward passes:

$$
\mathop{\mathbb{E}}_{\theta \sim q(\theta)} (f(e_l|S;\theta)) \approx \mu = \frac{1}{T} \sum_{t=1}^{T} f(e_l|S;\theta_t),
$$

$$
\mathop{\mathbb{V}\mathrm{ar}}_{\theta \sim q(\theta)} (f(e_l|S;\theta)) \approx \sigma^2 = \frac{1}{T-1} \sum_{t=1}^{T} (f(e_l|S;\theta_t) - \mu)^2,
$$

(9)

where $\theta_t$ denotes the model parameters for the $t$-th dropout sampling, and $T$ is the number of forward passes. A smaller variance indicates higher prediction confidence.

Afterward, we transform the inexact weakly-supervised learning task into a semi-supervised learning task, where high-confidence samples are treated as labeled samples, while the remaining samples are treated as unlabeled samples. For the abnormal sequence $S^+$, the top $r_{hi} \times N_{S+}$ behaviors with the smallest variance are selected as high-confidence samples, the next $r_{mid} \times N_{S+}$ behaviors are considered as medium-confidence samples, and the rest are treated as low-confidence samples:

$$
\Omega_{con}^{hi} = \mathrm{minTopK}_{e_l \in S^+}(\mathbb{V}\mathrm{ar}_{\theta \sim q(\theta)}(f(e_l|S^+;\theta)), r_{hi} \times N_{S+}),
$$

$$
\Omega_{con}^{mid} = \mathrm{minTopK}_{e_l \in S^+, l \notin \Omega_{con}^{hi}}(\mathbb{V}\mathrm{ar}_{\theta \sim q(\theta)}(f(e_l|S^+;\theta)), r_{mid} \times N_{S+}),
$$

(10)

where $\mathrm{minTopK}(\cdot, k)$ returns the indices corresponding to the $k$ smallest elements. For high-confidence samples, we utilize the expectation of their anomaly scores to generate hard pseudo labels for optimizing model parameters:

$$
\mathcal{L}_{hi} = \frac{1}{|\Omega_{con}^{hi}|} \sum_{l \in \Omega_{con}^{hi}} \mathrm{BCE}(f(e_l|S^+;\theta), \mathbb{1}(\mathbb{E}_{\theta \sim q(\theta)}(f(e_l|S^+;\theta)) > \tau_a)),
$$

(11)

where $\tau_a$ is the anomaly detection threshold, behaviors with anomaly scores greater than $\tau_a$ are considered as anomalies, behaviors with scores less than $\tau_a$ are considered normal, and $\mathbb{1}(\cdot)$ is an indicator function.

For those medium-confidence samples, directly using hard pseudo labels may introduce noise. To mitigate the impact of noise, we introduce more reliable soft pseudo labels to avoid high-confidence erroneous predictions by the model. We optimize the model by simultaneously considering less reliable hard pseudo labels $y_{hard}$ and more reliable soft pseudo labels $y_{soft}$ as follows:

$$
\mathcal{L}_{mid} = \frac{1}{|\Omega_{con}^{mid}|} \sum_{l \in \Omega_{con}^{mid}} \lambda_{pse} \mathrm{BCE}(f(e_l|S^+;\theta), y_{hard}) + (1 - \lambda_{pse}) \mathrm{BCE}(f(e_l|S^+;\theta), y_{soft}),
$$

(12)

where when $\mathbb{E}_{\theta \sim q(\theta)}(f(e_{hi}|S^+;\theta)) > \tau_c$, $y_{hard}$ is set to 1, and when $\mathbb{E}_{\theta \sim q(\theta)}(f(e_{hi}|S^+;\theta)) < 1 - \tau_c$, $y_{hard}$ is set to 0. $\tau_c$ is an adaptive threshold that increases with confidence and can be computed at the $t$-th iteration as $\tau_c^t = \beta_c \tau_c^{t-1} + (1 - \beta_c) \mathrm{maxNorm}\left(\mathbb{V}\mathrm{ar}_{\theta \sim q(\theta)}(f(e_k|S;\theta))^{-1}\right)$, with $\tau_c^0 = \tau_a$, where $\mathrm{maxNorm}(\cdot)$ is the maximum normalization operation. The soft pseudo label

324 $y_{soft} = f(e_l|S^+; \theta_{ema})$ is generated by the model's exponential moving average (EMA) model
325 acting as a teacher to guide the learning of the current model. The parameters $\theta^t_{ema}$ of the EMA
326 model at iteration $t$ can be computed as $\theta^t_{ema} = \beta_{ema}\theta^{t-1}_{ema} + (1 - \beta_{ema})\theta^t$. The total training loss
327 at this stage can be calculated as $\mathcal{L}_3 = \mathcal{L}_{hi} + \mathcal{L}_{mid}$.

Table 1: Performance comparison of RMSL with 16 baselines for behavior-level ITD. The best and second-best results are boldfaced and underlined, respectively. An upward arrow indicates the higher the better, and a downward arrow indicates the lower the better.

| Model | CERT r4.2 | | | | | | CERT r5.2 | | | | | |
|---|---|---|---|---|---|---|---|---|---|---|---|---|
| | AUC↑ | DR↑ | FPR↓ | DR@5%↑ | DR@10% | DR@15%↑ | AUC↑ | DR↑ | FPR↓ | DR@5%↑ | DR@10% | DR@15%↑ |
| DeepLog(Du et al., 2017) | 0.7469 | 0.7152 | 0.3767 | 0.2310 | 0.3842 | 0.4620 | 0.8549 | 0.7767 | 0.2336 | 0.4970 | 0.5954 | 0.6648 |
| TIRESIAS(Shen et al., 2018) | 0.8377 | 0.7820 | 0.2338 | 0.3761 | 0.5277 | 0.6484 | 0.8804 | 0.8129 | 0.2073 | 0.5463 | 0.6373 | 0.7297 |
| RNN(Elman, 1990) | 0.7521 | 0.6934 | 0.3622 | 0.2299 | 0.3821 | 0.4625 | 0.8641 | 0.8286 | 0.2361 | 0.4548 | 0.5928 | 0.6910 |
| GRU(Chung et al., 2014) | 0.7486 | 0.7119 | 0.3804 | 0.2391 | 0.3815 | 0.4614 | 0.8504 | 0.7911 | 0.2395 | 0.4637 | 0.5704 | 0.6499 |
| Transformer(Vaswani et al., 2017) | 0.7981 | 0.7195 | 0.2799 | 0.2918 | 0.4201 | 0.5321 | 0.8628 | 0.7621 | 0.1985 | 0.4858 | 0.5745 | 0.6694 |
| RWKV(Peng et al., 2023) | 0.8165 | 0.7923 | 0.2576 | 0.2630 | 0.4348 | 0.5886 | 0.8727 | 0.8020 | 0.2345 | 0.5380 | 0.6224 | 0.6887 |
| DIEN(Zhou et al., 2019) | 0.7894 | 0.7461 | 0.3072 | 0.4147 | 0.4875 | 0.5342 | 0.8268 | 0.7724 | 0.2690 | 0.3811 | 0.5455 | 0.6175 |
| BST(Chen et al., 2019) | 0.6777 | 0.6554 | 0.3451 | 0.1625 | 0.2614 | 0.3647 | 0.8162 | 0.7417 | 0.2301 | 0.4772 | 0.5650 | 0.6548 |
| FMLP(Zhou et al., 2022) | 0.8526 | 0.7983 | 0.2027 | 0.4783 | 0.5647 | 0.6837 | 0.8435 | 0.8757 | 0.2889 | 0.4278 | 0.5171 | 0.5659 |
| m-RNN | 0.8652 | 0.8032 | 0.1996 | 0.4375 | 0.6707 | 0.7549 | 0.9108 | 0.8131 | 0.1359 | 0.6881 | 0.7699 | 0.8230 |
| m-GRU | 0.8514 | 0.8103 | 0.2378 | 0.3364 | 0.5962 | 0.7001 | 0.9040 | 0.7879 | 0.1367 | 0.6780 | 0.7458 | 0.7957 |
| m-LSTM | 0.8531 | 0.7891 | 0.2259 | 0.3310 | 0.5908 | 0.6897 | 0.8985 | 0.7730 | 0.1385 | 0.6729 | 0.7329 | 0.7779 |
| m-Transformer | 0.8533 | 0.8005 | 0.2112 | 0.3109 | 0.5451 | 0.6951 | 0.8929 | 0.8358 | 0.1586 | 0.5684 | 0.7357 | 0.8247 |
| m-FMLP | 0.8837 | 0.8190 | 0.1671 | 0.4266 | 0.6772 | 0.7957 | 0.8920 | 0.8169 | 0.1614 | 0.4878 | 0.7412 | 0.8066 |
| ITDBERT(Huang et al., 2021) | 0.7413 | 0.6911 | 0.3153 | 0.2005 | 0.3272 | 0.4383 | 0.8139 | 0.6853 | 0.1996 | 0.5724 | 0.6243 | 0.6518 |
| OC4Seq(Wang et al., 2021) | 0.8113 | 0.8080 | 0.2940 | 0.1466 | 0.3019 | 0.4712 | 0.9202 | 0.8503 | 0.1727 | 0.6414 | 0.7383 | 0.8245 |
| **RMSL (Ours)** | **0.9701** | **0.9142** | **0.0924** | **0.7030** | **0.9245** | **0.9585** | **0.9568** | **0.8908** | **0.0950** | **0.7945** | **0.8645** | **0.9073** |
| Abs. Improv. | 0.0864 | 0.0952 | 0.0747 | 0.2247 | 0.2473 | 0.1628 | 0.0366 | 0.0151 | 0.0409 | 0.1064 | 0.0946 | 0.0826 |
| Rel. Improv.(%) | 9.78% | 11.62% | 44.70% | 46.98% | 36.52% | 20.46% | 3.98% | 1.72% | 30.10% | 15.46% | 12.29% | 10.02% |

## 4 EXPERIMENTS

**Experimental Settings**. sections A.1 to A.3 and A.5 describes detailed experimental information, including datasets, baseline methods, implementation details, and evaluation metrics.

**Overall Comparison**. Table 1 shows the performance of our RMSL and all the baseline methods on the behavior-level ITD task. The results indicate that our RMSL significantly outperforms existing baselines across all datasets on the behavior-level detection tasks. Specifically, in terms of the AUC metric, RMSL outperforms the best-performing baseline by 9.78% and 3.98% on the CERT r4.2 and r5.2 datasets, respectively; on the DR metric, it also outperforms by 11.62% and 1.72%, respectively; and on the FPR metric, it outperforms by 44.70% and 30.10%, respectively. Previous baseline methods all considered how to better model normal behavior patterns, whether by designing better structures to describe behavior features, or designing different tasks to learn normal behavior patterns such as next behavior prediction (Deeplog (Du et al., 2017), TIRESIAS (Shen et al., 2018), RNN (Elman, 1990), GRU (Chung et al., 2014), Transformer (Vaswani et al., 2017), RWKV (Peng et al., 2023), DIEN (Zhou et al., 2019), BST (Chen et al., 2019), FMLP (Zhou et al., 2022)), masked behavior prediction (m-RNN, m-GRU, m-LSTM, m-Transformer, m-FMLP), or one-class classification based on minimizing hyper-spheres (OC4Seq(Wang et al., 2021)). Without any prior information about anomalies, the performance of these approaches has reached a bottleneck. Analyzing the possible reasons, the paradigm of simply treating deviations from normal as anomalies is inappropriate, as these models cannot truly distinguish between normal and abnormal. Since training sets cannot encompass all normal behavior patterns in the real world, these methods might misclassify unseen but normal behavior patterns as anomalies, leading to a high false positive rate. Reflecting on the field of cybersecurity, there may be another issue where some malicious users often disguise themselves as normal users to perform subtle malicious behaviors, making them very difficult to distinguish and leading to a low detection rate. ITDBERT(Huang et al., 2021) is a sequence-level supervised method, but it can also indirectly provide behavior-level scores by interpreting the model's predictions using attention scores. However, its performance lags significantly behind our method. In this work, we conducted a more practical weakly supervised setting, and experiments proved that this led to significant performance gains. Furthermore, our method starts with a one-class classification model and gradually enhances its behavior-level classification capabilities by introducing sequence-level labels, making it more flexible.

**Ablation Study**. The final RMSL is primarily trained in three training stages: multiple hyper-spheres based zero positive warm-up, multiple instance learning, and adaptive behavior-level self-training debiasing. To better understand how different training stages contribute to the final performance, we conducted ablation studies. Specifically, we introduced four variants of RMSL, each corresponding to models trained with different combinations of stages.

Figure 3 shows the results of the ablation study. The experimental results show that the model trained using the first two stages (**"stage 1+2"**) achieved significant improvements across all metrics compared to the model trained only with the first stage (**"stage 1"**), demonstrating the effectiveness of the second training stage multiple instance learning. The model trained using all three stages (**"stage 1+2+3"**) outperformed other variants, showing a further slight improvement

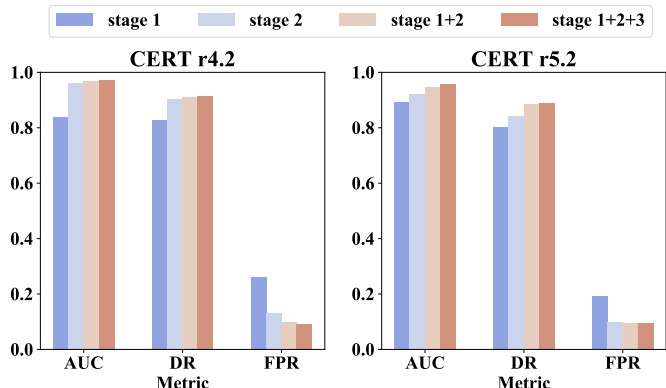

Figure 3: Results of the ablation study.

in metrics compared to **"stage 1+2"**, which confirms the effectiveness of the third training stage adaptive behavior-level self-training debiasing. Additionally, the metrics of **"stage 1+2"** were better than the model trained solely with the second stage (**"stage 2"**), proving the effectiveness of the first training stage multiple hyper-spheres based zero positive warm-up. A warm start from a one-class model, which can provide initial anomaly score predictions, can help MIL optimize better.

**Hyper-parameter Analysis**. In this subsection, we analyze the impact of two key hyper-parameters on the performance of RMSL across the CERT r4.2 and r5.2 datasets. Firstly, as depicted in Figure 4a, we varied the number of hyper-spheres based normal prototypes $M$ from 1 to 100 (step size 10). Observations indicate that with the increase of $M$, the AUC initially increases and then slightly decreases on both datasets, achieving the optimal performance when $M$ is set to 40.

The initial increase suggests that using multiple hyper-spheres as prototypes to represent normal patterns of behaviors is more expressive than compressing all normal behaviors into a single minimum volume hyper-sphere in the latent space (similar to Deep SVDD(Ruff et al., 2018)). The subsequent decline may be due to redundancy when the number of hyper-spheres exceeds the number of normal patterns, in-

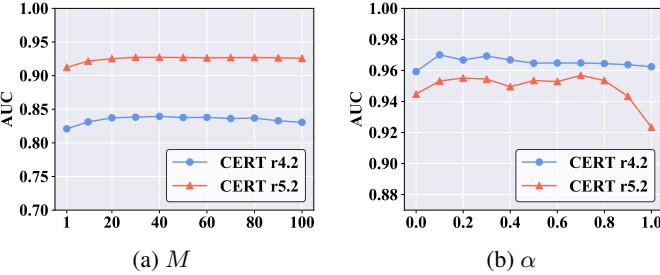

Figure 4: The influence of number of hyper-spheres based normal prototypes $M$ and dual scoring balance factor $\alpha$.

creasing the likelihood that individual anomaly behaviors are incorrectly assigned to one of the hyper-spheres. In Figure 4b, we tuned the dual scoring balance factor $\alpha \in [0, 1]$ (step size 0.1), which balances the contributions of the discriminative score and the hyper-sphere-based deviation score. Setting $\alpha = 0$ relies solely on the deviation score, whereas $\alpha = 1$ uses only the discriminative score. Optimal performance is achieved at $\alpha = 0.1$ (r4.2) and $\alpha = 0.6$ (r5.2), highlighting dataset-specific trade-offs between the two scoring mechanisms.

**Visualization**. We also conducted visualization experiments to compare the embedding vectors between the zero-positive setting ( Figure 5a ) and the weak supervision setting ( Figure 5b ). Visualizations show red/black dots for anomalous/normal behaviors and blue hypersphere centers. The zero-positive setting exhibits significant embedding vector overlap, whereas the weak supervision setting achieves clear separation. For more detailed information, please refer to section A.6.

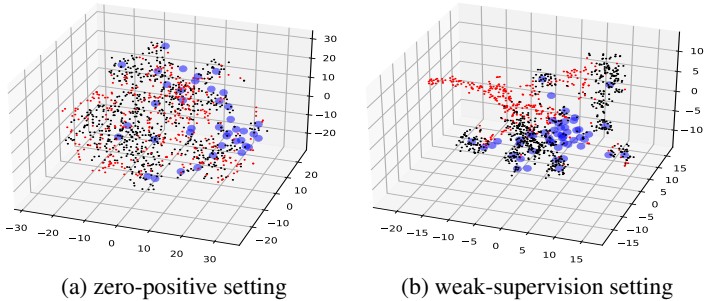

(a) zero-positive setting      (b) weak-supervision setting

Figure 5: Embedding vectors visualization

## 5 RELATED WORKS

### 5.1 INSIDER THREAT DETECTION

In recent years, numerous studies have explored the application of deep learning techniques in Insider Threat Detection (ITD). These works treat user activities as behaviors and aggregate them into sequences, then leverage sequence models from the field of Natural Language Processing (NLP) to capture the temporal dependencies between user activities for anomaly detection (Tuor et al., 2017; Yuan et al., 2019; Vinay et al., 2022; Huang et al., 2021; Yuan et al., 2020; Lu & Wong, 2019). Yuan et al. (2019) proposed a model that combines temporal point processes and recurrent neural networks to capture temporal information and activity types within sessions for sequence-level ITD. Furthermore, their subsequent work(Yuan et al., 2020) proposed a framework that combines metric-based few-shot learning and self-supervised pre-training methods to discover new malicious sessions and detect insiders through similarity scores. Huang et al. (2021) pre-trained a language model BERT(Devlin et al., 2018) on historical activity data to capture fused semantic representations and proposed an attention-based architecture to detect malicious activities of compromised internal nodes within a network. Tuor et al. (2017) use daily features for each user as historical feature vectors to predict the feature vectors of the next day, thereby enabling the detection of day-level insider threats. However, these methods can only determine whether a sequence is anomalous but fail to detect specific anomalous behaviors.

### 5.2 ANOMALY DETECTION WITH INEXACT SUPERVISION

Anomaly detection with inexact supervision refers to effectively identifying anomalies using coarse-grained labels. Current research mainly focuses on video anomaly detection tasks. Sultani et al. (2018) is the first to formulate anomaly detection with weakly supervised video-level labels as a Multiple Instance Learning (MIL) problem, treating each video as a bag of instances and using video-level anomaly labels to learn the anomaly scores of individual video segments. Tian et al. (2021) trained a classifier using the top K instances with the highest anomaly scores to learn more robust temporal features for identifying abnormal segments. Chen et al. (2023) proposed a feature magnitude contrastive loss to address the issue where the magnitude of normal instances is greater than that of anomalous instances due to changes in scene attributes, thereby enhancing the separability between normal and anomalous features. Differently, Lv et al. (2023) identified the problem of biased sample selection in MIL and proposed an unbiased MIL framework to enhance the detector's ability to distinguish between normal and anomalous behaviors, eliminating selection bias. To further improve the performance of weakly supervised video anomaly detection models, other studies have focused on applying two-stage training schemes. Specifically, Feng et al. (2021) introduced a self-training framework based on MIL, using a pseudo-label generator and a self-guided attention encoder to improve anomaly detection performance. Li et al. (2022) enhanced the MIL framework by improving sample selection, proposing multi-sequence learning to select consecutive segments with high anomaly scores. Although MIL methods have been widely applied in video anomaly detection, research in insider threat detection has not been fully explored.

## 6 CONCLUSIONS

In this paper, we propose a novel weakly supervised learning framework, Robust Multi-sphere Learning (RMSL), to address the challenge of sparse behavior-level labels in fine-grained ITD. This method models diverse normal behavior patterns through multiple hyper-spheres and determines anomalies by combining classification separability with the degree of deviation from the hyper-spheres. We adopt a three-stage progressive training strategy to obtain RMSL: first, a multi-sphere-based one-class model is trained in the zero positive scenario. Then, sequence-level weak labels are introduced to further optimize the model and enhance its ability to distinguish between normal and anomalous behaviors. Finally, a debiasing technique is applied to eliminate prediction bias. Experimental results show that RMSL significantly outperforms existing methods in insider threat detection tasks. However, our approach still has limitations. Although weak labels reduce the annotation cost, their quality (e.g., whether the entire behavioral sequence is accurately labeled as normal or abnormal) may also affect model performance. Future research will further focus on the evaluation and optimization of weak label quality to further enhance the practicality of this method.

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

# A APPENDIX

## A.1 DATASETS

To evaluate the performance of our approach on the behavior-level ITD task, following previous studies (Liu et al., 2019; Li et al., 2023; Yuan et al., 2020; AlSlaiman et al., 2023; Le et al., 2020; Yuan & Wu, 2021; Xiao et al., 2022), we selected two publicly available datasets, **CERT r4.2** and **CERT r5.2** (Lindauer, 2020), which correspond to detection scenarios with different data scales and are widely used in the field of insider threat detection. These datasets encompass a variety of user behavior categories including logon/logoff, email communications, file accesses, device operations, and HTTP requests, each associated with a timestamp. For both CERT r4.2 and CERT r5.2 datasets, we aggregated user log data from multiple sources in chronological order and appended each user's behaviors to their historical behavior sequences. Sessions were defined using "login" and "logout" behaviors as delimiters, thereby dividing the data into individual sessions, each treated as a behavior sequence. Given that both datasets cover a period of one and a half years, we utilized the first year's data for model training and validation, while reserving the remaining six months' data for performance evaluation. For the training set, we only utilized sequence-level labels to optimize the model, whereas for the test set, we used behavior-level labels to evaluate performance. Detailed information about the datasets is summarized in Table 2.

Table 2: Statistics of the datasets.

| Dataset | CERT r4.2 | CERT r5.2 |
|---|---|---|
| # Normal Sequences | 469,478 | 1,004,791 |
| # Abnormal Sequences | 1134 | 1843 |
| **Seq.-level Imb. Ratio** | **414** | **545** |
| # Normal Behaviors | 32,762,906 | 79,846,358 |
| # Abnormal Behaviors | 7,316 | 10,306 |
| **Beh.-level Imb. Ratio** | **4,478** | **7,748** |

## A.2 BASELINES

To better demonstrate the performance of our RMSL model, we compared it with 16 state-of-the-art baselines. Note that, since our evaluation granularity is at the behavior level, we only considered methods that can perform behavior-level anomaly detection without relying on behavior-level annotations. DeepLog(Du et al., 2017) and TIRESIAS(Shen et al., 2018) are two classic methods that fit normal behavior sequences by learning to predict the next behavior given the context of the behavior sequence. They can detect anomalies by determining if each input behavior deviates from the model's prediction. The backbone of DeepLog is a two-layer stacked LSTM(Hochreiter & Schmidhuber, 1997), whereas TIRESIAS maintains a single-layer LSTM but improves upon DeepLog by constructing a more complex memory structure within the LSTM unit. In addition to LSTM, we also tried three other widely popular sequence modeling architectures RNN(Elman, 1990), GRU(Chung et al., 2014), Transformer(Vaswani et al., 2017), and RWKV(Peng et al., 2023) to learn to predict the next behavior and detect anomalies. Furthermore, we compared two classic user behavior modeling methods, DIEN(Zhou et al., 2019) and BST(Chen et al., 2019), which can predict the probability of user behaviors occurring, and a recent method, FMLP(Zhou et al., 2022), which filters noise from historical user behavior data to predict future user behaviors. The aforementioned models only consider the context before the occurrence of a behavior when predicting whether a behavior is abnormal. To fully utilize the context information of the entire behavior sequence, we allowed the models to access the entire behavior sequence and constructed a masked behavior prediction task, similar to LogBERT(Guo et al., 2021). In this task, a specific behavior in the behavior sequence is replaced with a mask identifier. We used bidirectional RNN, GRU, and LSTM, as well as Transformer and FMLP, to learn to predict the behavior at the masked position. Anomalies are detected by determining if each masked behavior deviates from the model's prediction, and these methods are referred to as m-RNN, m-GRU, m-LSTM, m-Transformer, and m-FMLP, respectively. ITDBERT(Huang et al., 2021) is an attention-based behavior-level detection method. The attention weights reflect the contribution of each behavior in the behavior sequence to predicting whether

the entire sequence is abnormal, allowing for the detection of abnormal behaviors based on these attention weights. Lastly, we also compared a representative method for treating ITD as a one-class classification problem, OC4Seq(Wang et al., 2021). This method learns to embed normal behaviors into a hyper-sphere, detecting anomalies by predicting how close behaviors are to the center of the hyper-sphere.

### A.3 IMPLEMENTATION

Our RMSL method is trained using the AdamW (Loshchilov & Hutter, 2017) optimizer with a weight decay of 0.0005. During the first stage of model training, the initial learning rate is set to 2e-6, during the second stage, the learning rate is set to 1e-5, and during the third stage, the learning rate is set to 1e-6. The batch size is set to 128, with each mini-batch consisting of 64 randomly selected normal sequences and 64 abnormal sequences. For the dual scoring balance factor $\alpha$, we set it to 0.1 for the CERT r4.2 dataset and 0.7 for the CERT r5.2 dataset respectively. Regarding the number of hyper-spheres $M$, we set it to 40 and randomly initialized each hyper-sphere center. The rationale for selecting these two key parameters is reported in the hyper-parameter analysis part of Section 4. We adopt the grid search strategy and leverage hyperparameter tuning tools[1] to achieve optimal performance, such as setting the hyper-sphere separability loss $\lambda_{sep} = 0.5$. For all experiments, for a fair comparison, our method is set with the same embedding size of 128 as all baseline methods and is trained for 10 epochs using an early stopping strategy. The experiments were conducted on a server with 2 Intel Xeon Gold 6226R CPUs running at 2.90GHz, 256GB of RAM, and one A6000 GPU with 48GB memory. The toolkit used for the experiments included Python 3.8, PyTorch 1.13.

### A.4 COMPUTATIONAL PERFORMANCE

We evaluated the computational performance of our model on a single NVIDIA A6000 GPU. For CERT 4.2, training on 32M behaviors completed in under 50 minutes, and inference on 1.9M behaviors required only 9 seconds, corresponding to an average latency of 0.0046 ms per behavior. For CERT 5.2, training on 79M behaviors completed in under 100 minutes, while inference on 7.5M behaviors took 27 seconds, yielding an average latency of 0.0036 ms per behavior. GPU memory usage scaled predictably with batch size, ranging from 2,661 MB at a batch size of 128 to 8,768 MB at 512. These results demonstrate that the model can be efficiently trained on large-scale behavioral datasets within practical time limits while supporting fast and memory-efficient inference, highlighting its scalability and suitability for real-world deployment.

### A.5 METRICS

Similar to the previous works (Buczak & Guven, 2015; Le et al., 2020; Le & Zincir-Heywood, 2021a; King & Huang, 2023; Cai et al., 2024), we use the behavior-level area under the ROC curve (AUC), detection rate (DR), and false positive rate (FPR) as evaluation metrics for all datasets and models. Here, DR = TP/(TP + FN) and FPR = FP/(FP + TN). TP, FN, FP, and TN represent the number of true positives, false negatives, false positives, and true negatives, respectively. Furthermore, following previous studies (Le & Zincir-Heywood, 2021a;b; Tuor et al., 2017; Cai et al., 2024), we also report the detection rates DR@5%, DR@10%, and DR@15% under investigation budgets of 5%, 10%, and 15% of the total number of behaviors, respectively.

### A.6 VISUALIZATION DETAILS

We visualized the learned embedding vectors using t-SNE (Van der Maaten & Hinton, 2008) and compared between the zero positive setting and the weak supervision setting, as shown in Figure 6, where red dots represent anomalous behaviors, black dots represent normal behaviors, and blue markers indicate centers of hyper-spheres. Each setting displays 3D projections from two different angles. Figure 6a demonstrates the embedding vectors produced by the one-class model trained with the first training stage multiple hyper-spheres based zero positive warm-up, which only uses normal sequences for training. It can be observed that in the latent space, dots representing normal behaviors and anomalous behaviors cannot be well separated, indicating poor inter-class separability. Figure 6b presents the embedding visualization results after further introducing sequence-level weak supervision

---

[1]https://github.com/microsoft/nni

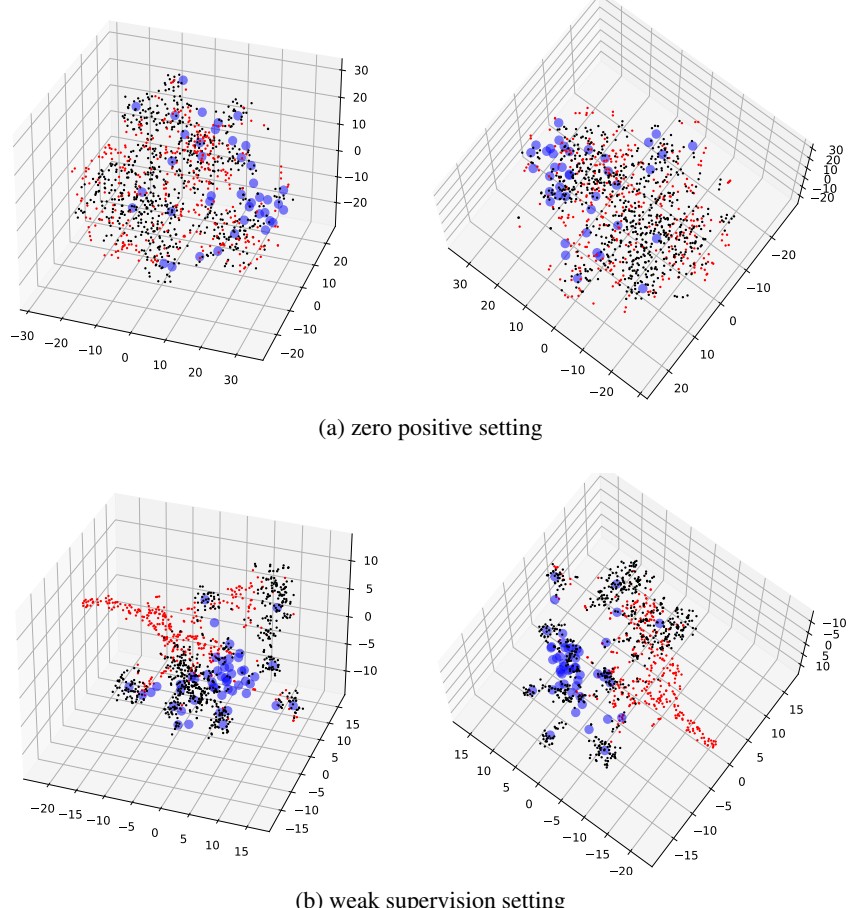

(a) zero positive setting

(b) weak supervision setting

Figure 6: Visualization of the embedding vectors of RMSL in the zero positive setting and the weak supervision setting.

signals based on the one-class model. From the figure, it can be seen that normal behaviors tightly cluster around their respective hyper-sphere centers and maintain a clearer separation from anomalous behaviors, reflecting the optimization of the decision boundary between normal and anomalous patterns using weak supervision signals, which effectively enhances inter-class distinguishability.

## A.7 ETHICS STATEMENT

This research does not involve human or animal subjects. All datasets utilized are publicly available and do not contain sensitive or personally identifiable information. Our work does not present potential risks related to harmful applications, such as discrimination, bias, or security concerns. The authors have no conflicts of interest or financial relationships relevant to this study. All experiments and analyses were conducted with integrity, transparency, and in accordance with ethical research practices.

## A.8 REPRODUCIBILITY STATEMENT

We have made every effort to ensure the reproducibility of our results. In this study, we propose a novel model. The datasets and preprocessing methods used for training and evaluation are detailed in section A.1, the baseline models for comparison are listed in section A.2, implementation details including hyperparameter settings, optimizer types, and runtime environment are provided in section A.3, and the evaluation metrics are described in section A.5. Furthermore, the supplementary

materials include code that can be used to reproduce the experiments, ensuring transparency and facilitating verification of our findings.

### A.9    DECLARATION OF LLM USAGE

The core methodological development in this study did not involve the use of large language models (LLMs) as any essential, original, or non-standard component. LLMs were employed only minimally, for purposes such as text polishing and language refinement.

