# OpenReview forum: "RMSL: Weakly-Supervised Insider Threat Detection with Robust Multi-sphere Learning"
_ICLR.cc/2026/Conference — ICLR 2026 Conference Withdrawn Submission_

### Official Review · Reviewer_5D24 · 2025-10-23

**Soundness:** 3
**Presentation:** 3
**Contribution:** 3
**Rating:** 4
**Confidence:** 3

**Summary:**

This paper presents RMSL (Robust Multi-Sphere Learning), a weakly-supervised approach for insider threat detection that uses sequence-level annotations to perform behavior-level anomaly detection. RMSL is a combination of techniques that operate in a sequence, i) multiple hyper-spheres, ii) one-class classifier, iii)multiple instance learning, followed by iv) anomoly detection. The method employs multiple hyper-spheres to model normal behavior patterns and implements a three-stage progressive training strategy to address the challenge of limited behavior-level annotations in cybersecurity. The paper claims the experiments significantly improve the performance over the SOTA methods.

**Strengths:**

- Novel multi-sphere approach that better captures diverse normal behavior patterns compared to single-sphere methods
- Well-designed three-stage progressive training strategy that systematically addresses different aspects of the learning problem
- Dual-scoring mechanism combining classification separability and deviation from normal patterns provides comprehensive anomaly assessment

**Weaknesses:**

- The paper is somewhat difficult to read through with a lot of compound sentences in general. It would have been an easier job for the reader has there been a better and simple and sentences with active voice. However, was able to grasp atleast some of the ideas presented in the paper.
- Major concern: Only evaluated on two datasets from the same source (CERT r4.2 and r5.2), raising serious questions about generalizability
- Parameter sensitivity issues: different optimal \alpha values needed for different datasets suggests the method may not be robust
- Computational complexity not adequately discussed - three-stage training likely increases computational burden significantly

**Questions:**

- Can you add the necessary clarifications on some of the terminology? Useful for non-experts in the domain and adds better clarity to the whole paper.
- Can you provide evaluation on datasets from different domains (network intrusion, fraud detection) or different insider threat datasets beyond CERT to demonstrate broader applicability?
- How can you determine optimal hyperparameters (especially \alpha) for new datasets without extensive tuning? Are there any principled guidelines for setting these hyper params and do they really have to be so different for each dataset?
- What is the computational complexity and training time compared to baselines? How does this scale with dataset size and number of spheres?
- How does performance degrade with varying levels of noise in sequence-level labels? What is the minimum label quality required for effective performance?

---

### Official Review · Reviewer_3uUr · 2025-10-27

**Soundness:** 3
**Presentation:** 3
**Contribution:** 3
**Rating:** 4
**Confidence:** 1

**Summary:**

This paper addresses the challenge of behavior-level anomaly detection in insider threat detection, where fine-grained labels are unavailable. To overcome the high cost of annotation and the limitations of unsupervised methods, the authors introduce weak sequence-level labels.
They propose a novel framework, Robust Multi-sphere Learning (RMSL), which uses multiple hyper-spheres to model normal behavior patterns. The framework is built in two key stages:
1.It first constructs a one-class classifier without any anomaly supervision as a starting point.
2.It then refines the model using the weak labels by integrating multiple instance learning and an adaptive behavior-level self-training debiasing mechanism based on prediction confidence.
3.This process enhances feature discrimination. Extensive experiments show that RMSL significantly improves performance for behavior-level insider threat detection.

**Strengths:**

This work's primary strength is its pragmatic and cost-effective approach to a complex problem. By leveraging weak sequence-level labels instead of costly fine-grained annotations, it offers a feasible solution for real-world deployment.
The proposed RMSL framework is methodologically robust, combining a strong, unsupervised starting point (a one-class classifier) with a sophisticated refinement process using multiple instance learning and adaptive self-training debiasing. This hybrid design effectively bridges the gap between fully unsupervised and fully supervised methods.
Ultimately, its key strength is demonstrated by significantly improving detection performance for fine-grained, behavior-level anomalies, a task where previous methods struggled with high false positive and miss rates.

**Weaknesses:**

1.Dependence on weak label quality: The framework's performance is inherently tied to the quality and representativeness of the provided weak sequence-level labels. Noisy or biased weak labels could significantly degrade performance, a vulnerability not discussed.
2.Limited interpretability: The use of multiple hyper-spheres and a complex refinement process may result in a "black-box" model. The method likely lacks clear interpretability for why a specific behavior is flagged as anomalous, which is crucial for security analysts.
3.Hyperparameter snsitivity: Methods involving multiple instance learning and self-training often have numerous hyperparameters (e.g., number of spheres, confidence thresholds). The model's performance is probably sensitive to these settings, but their tuning and the associated robustness are not elaborated upon.
4.Scalability to massive log data: While effective, the framework's scalability to truly massive, enterprise-scale log data with millions of events is not demonstrated. The computational cost of the multi-sphere representation on such data remains an open question.

**Questions:**

In fact, I’m not familiar with this research topic. This work appears to be quite comprehensive and is well-written. My primary concern is that the references are out of date, e.g. before 2021. Same issues with the compared baselines.

---

### Official Review · Reviewer_ubTb · 2025-10-29

**Soundness:** 2
**Presentation:** 3
**Contribution:** 2
**Rating:** 4
**Confidence:** 4

**Summary:**

The paper introduces an algorithm to detect insider threats using weak-supervision. It starts with an simple unsupervised discriminative model for detecting anomalies and then progresses through stages of complexity. In stage 1, the composite model M comprising of a multi-hypersphere model and an anomaly classifier is trained unsupervised. In stage 2, the algorithm uses a supervised multiple instance learning algorithm to fine-tune the model M. In stage 3, the model is further fine-tuned in a semi-supervised manner with pseudo-labels. The pseudo-labels (normal/anomaly) are assigned to instances on the basis of their anomaly scores and their variances (which act as proxy for confidence).

**Strengths:**

The paper is well-written and the reasoning behind the algorithmic decisions are sound. The set of ablation experiments is very good.

**Weaknesses:**

The main weakness of the paper is in the experiment methodology. These (among other comments) are elaborated in more details in the main comments below.

1. Equation 5: It appears that 'dual-scoring' is just another name for 'ensemble' (with two members). This is not very novel. We could even generalize to more than two to include more 'complementary perspectives'.


2. Line 220/221 Multi-Center loss: How was the number of hyper-spheres determined in the experiments? Is that auto determined or a hyper-parameter? From line 401 it seems that this parameter was set after doing the ablation study on the entire dataset. This implies that the 'M' parameter was set unfairly, i.e., seeing which number gives the best results and then reporting those best results.


3. Equation 8: Should this be average on both sides, instead of just summation? Else the inequality might be biased towards the category (+/-) whichever has more labels.


4. Section 3.4: In stage 3, are the pseudo labels computed only once? A more natural approach might be to do this in a E-M (expectation-maximization) style where: 1. the pseudo labels are computed, 2. then the model is fine-tuned, 3. go back to step 1 to recompute the pseudo labels; and repeat this cycle for a few iterations till some stability is attained through label-diffusion.


5. Figure 3: Adding stage 3 (stage 1+2+3) does not seem to add much benefit. Only two datasets were used for the evaluation and the benefit of stage 3 looks highly speculative.


6. Lines 417--419 and 767--768: Clearly, the hyper-parameter values were set after-the-fact. There should be a more automatic way to set the \alpha parameter (maybe use a separate independent dev/validation set).

**Questions:**

1. Section 3.4: In stage 3, are the pseudo labels computed only once during the entire training?

2. Could we set the hyper parameters (number of hyper-spheres and \alpha) in a more automatic manner or using an independent dev set that does not raise questions about fairness?

---

### Official Review · Reviewer_Ag8f · 2025-11-04

**Soundness:** 3
**Presentation:** 4
**Contribution:** 3
**Rating:** 6
**Confidence:** 2

**Summary:**

Much existing work in insider threat detection focuses on unsupervised methods aimed at learning what constitutes "normal" behavior.  However, this work tends to assume there is a single type of "normal" behavior, limiting the expressiveness of such representations.  Existing work has also largely focused on sequence-level anomaly detection.  While data is more available at a sequence level, it can be beneficial to detect actual specific anomalous behaviors within potentially-long sequences.  To address these deficiencies, the authors propose RMSL.  RMSL combines a multiple-hyper-sphere representation of "normal" behavior with a multiple instance learning-based classifier that detects anomalous behaviors with access to only labeled full sequences.  The authors define the pieces of RMSL and its multi-stage training process and empirically evaluate it on two CERT datasets, which are popular benchmarks in ITD.

**Strengths:**

I think this is an interesting paper.  The behavior-level anomaly detection focus is interesting, and I think a situation where only sequence-level labels are available is realistic and well-motivated.  The combination of normal-behavior modeling and multiple instance learning is clever and well-designed, and I think the authors' description is clear and well-laid out.  Overall, I found the paper well-written and compelling.

**Weaknesses:**

Figure 1 is described as showing the setting you aim to address (WITD), but it's not ever really described.  From looking at Figure 1, it seems as though the marked "ambiguous anomaly interval" is characterized by a decrease in behavior, as seen by the gap in the sequence of "behavior" page images to the left of it.  This suggests that "behaviors" are some sort of temporally-annotated events.  However, Section 2 makes it clear that time isn't being explicitly represented in behavior sequences, since each behavior in the sequence is only indexed by l (its count in the sequence).  It's also not entirely clear what the term "ambiguous anomaly interval" means, since that phrase isn't used elsewhere in the paper.

In the contribution list, the first contribution is essentially "We create an algorithm called RMSL", and the second contribution is "we propose an anomaly detector with a three-stage training process" - which is to say, RMSL.  So both the first and second contributions seem to be "We propose RMSL".  If this isn't the case, the contributions should be reworked to be clearer.

While you define a sequence and a behavior in Section 2, it would be helpful to give an example and define the space of behaviors.

On a similar note, along with examples, it would be helpful to discuss if there are any assumptions you are making about the types of anomalies you're detecting and the types of sequences you're considering.  This especially came to mind in Section 3.3.  You state that, adopting the MIL framework, you "consider the sum of the anomaly scores of the highest-scoring behaviors within a sequence as the anomaly score for the entire sequence."  How do you define the "highest-scoring behaviors"?  Is there a score threshold? Do you only pick multiple when there's a tie for "highest-scoring"?  Or are you always picking the "top n" anomalous behaviors in a sequence?  Depending on the answer to that, I would imagine different types of behavioral anomalies are expected (e.g., a couple of sequential anomalous behaviors in an otherwise-normal sequence vs a sequence where all behaviors are slightly anomalous).

While I was able to follow pretty well up through page 5, Section 3.4 and the description of the self-training debiasing stage lost me a bit.  By the time we get to Section 3.4, the reader has more context for the rest of RMSL, so a few additional sentences at the top of the section giving higher-level intuiting would help here.

The experiments and description around them are the weakest part of the paper for me.  While I understand leaving some of the experiment details in the appendix due to space limitations, the complete lack of detail in the main body of the paper in the current version doesn't give the reader enough context to reasonably interpret the experimental results.  At the very least, given that you haven't given examples previously about the types of events and sequences that you expect for RMSL, you need to describe the CERT datasets - how long are the sequences, what is the behavior space (how many types of behaviors do you consider), what percentage of behaviors are anomalies, what do those anomalies look like (e.g., are they singleton anomalous behaviors, or are they groups of sequential anomalous behaviors), etc.  Even the description in the appendix right now doesn't answer all of these questions for me (and I'm not sure, in Table 2 in Appendix A.1, what "Imb." stands for, since it's not defined or discussed in the paragraph from what I can find).

It would also be helpful to discuss what types of comparison methods you're evaluating in Table 1.  It sounds like many of them are simply modeling the normal behavior, which is a fair baseline, but it's unclear which, if any, of the comparison methods are able to make use of the labelled ground-truth sequences.  Especially for CERT r4.2, Figure 3 suggests that you can do better than basically all other methods by simply doing the stage 2 piece, which is the part that trains against labeled sequences.  I understand that, if the current literature either does only sequence-level ITD or fully unsupervised, that there may not be fully equivalent methods to compare against.  However, if that's the case, that should be discussed and made clearer in Section 4.

**Questions:**

When doing the first training stage (the zero-positive initialization), do you assume that there are no anomalous behaviors in the training data, or simply that any anomalous behaviors are rare enough to not overly bias the sphere estimation?

Which of the methods compared against in Table 1 make use of the sequence-level labels?

---

### Note · Authors · 2025-12-04

**Comment:**

Dear Editors and Reviewers,

After carefully reviewing the valuable feedback provided by the reviewers, we have come to the conclusion that it is necessary to withdraw our submission at this time. We deeply appreciate the insightful comments and constructive suggestions from all the reviewers, which have been instrumental in helping us identify areas where our work can be significantly improved.

Based on the feedback received, we plan to take the following steps to strengthen our manuscript:

Conduct additional experiments to further validate our results and address key questions raised by the reviewers.
Perform a more comprehensive analysis of certain critical hyperparameters to ensure robustness and reproducibility.
Provide more detailed explanations of our experimental setup and methodology to improve clarity and transparency.
Refine the presentation of our ideas and revise ambiguous or unclear statements to enhance the overall quality of the manuscript.
However, after careful consideration, we believe that it will not be possible to complete all the necessary supplementary experiments and analyses within the limited time available. To ensure the completeness and scientific rigor of our study, we require additional time to fully refine the manuscript. Therefore, we have decided to temporarily withdraw our current submission and plan to resubmit after completing all the necessary improvements.

We believe that these improvements are essential to make our work more solid and impactful before resubmitting it for publication. While this process will require additional time and effort, we are committed to ensuring that our research meets the highest standards of rigor and quality.

We sincerely apologize for any inconvenience caused by this withdrawal and are extremely grateful for the time and effort invested by the editors and reviewers in evaluating our work. Your feedback has been invaluable in guiding us toward a stronger version of this manuscript.

**Withdrawal Confirmation:**

I have read and agree with the venue's withdrawal policy on behalf of myself and my co-authors.